# Impact of online negative and positive reviews on app innovation

Lin Sun⊕, Yuting He, Feng Fu⊕*

School of Business, Chengdu University of Technology, Chengdu, P.R. China

⊕ These authors contributed equally to this work.
* fufeng@cdut.edu.cn

## Abstract

Past studies have suggested that online reviews positively impact app innovation. However, extant research has not yet explored the distinct impacts of online negative and positive reviews on app innovation. Based on signaling theory and negative bias, this study empirically examines the effects of online negative reviews versus online positive reviews on app innovation by using panel data from the iOS App Store in China. The findings demonstrate that online negative reviews have a more positive influence on app innovation than online positive reviews. Additionally, compared with online positive reviews, app performance more effectively weakens the promoting effect of online negative reviews on app innovation. Moreover, both app history and platform owner's entry play a positive moderating role in the impact of online negative reviews on app innovation, while no positive moderating effect is observed in the impact of online positive reviews on app innovation. These results demonstrate the different effects of online negative reviews and online positive reviews on app innovation, expand the contingent value of online reviews and app innovation.

## 1. Introduction

With mobile devices and the Internet now easily accessible worldwide, the popularity of mobile applications (apps) has increased. However, despite the overall growth of the mobile app industry, individual app success rates remain low. For example, ~50% of iOS apps generate < $500 in monthly revenue, while only ~0.1% of apps in the Android ecosystem surpassed 5 million downloads as of the first quarter of 2018 [1]. To survive in the hypercompetitive app market, developers often need to introduce new functionalities or features frequently, typically releasing new versions on a biweekly or monthly basis [2]. Therefore, effectively gathering user demand information and fostering app innovation has become a key concern for both enterprises and academics.

Previous research has shown that online reviews serve as a vital channel for app developers to effectively gather information about users' needs. Some studies have

**Data availability statement:** All relevant data for this study are publicly available from the figshare repository (https://figshare.com/articles/dataset/raw-data_xlsx/28746656).

**Funding:** This work was financed by the Philosophy and Social Science Research Fund of Chengdu University of Technology (YJ2024-QN012 to F.F.) and the Philosophy and Social Science Planning Project of Chengdu (2024BS071 to F.F.)

**Competing interests:** No authors have competing interests.

observed that online reviews, as an information resource, can significantly promote app innovation [2,3]. According to signaling theory, review information—such as user needs and satisfaction—acts as a signal that reduces information asymmetry between users and developers, which in turn influences developers' innovation performance [4]. However, these studies overlook the distinct emotions conveyed in online reviews, specifically negative and positive reviews. According to negative bias theory, the usefulness and value of online negative and positive reviews vary [5,6]. Specifically, users tend to pay more attention to negative reviews because, compared with positive reviews, negative reviews provide greater value, contain more detailed user feedback on product quality, and help other users better evaluate the quality of a product [7,8]. Consequently, users' purchasing decisions are more likely to be influenced by online negative reviews [9,10]. Given the heterogeneity in usefulness and value between online negative and positive reviews, an important question arises: Do app developers differ in how they perceive and utilise these review types? If so, do these differences result in distinct impacts on app innovation? However, limited research has explored these inquiries.

Additionally, because app operations depend on third-party platforms such as Apple or Google, innovation is influenced by the app resources and by the competitive dynamics with the platform owner. This interplay creates a complex set of contingencies that shape the impact of users' online reviews on app innovation. Specifically, app innovation is a knowledge management process in which developers utilise their resources and capabilities to identify, assimilate, and integrate external knowledge. The varied history and performance of apps indicate that developers vary substantially in both resource appropriation and management capabilities. Furthermore, the mechanism through which online negative and positive reviews impact app innovation is distinctive.

Second, as leaders of the ecosystem, platform owners are responsible for collaborating with developers to have them enhance complementary innovations that add value to the platform ecosystem and its partners [11]. The platform owner often aims to capture substantial potential value or provide motivation for complementary innovations by introducing competition [12]. Platform owners usually imitate their complementors' products with similar offerings [13], which in turn positions them as competitors against their complementors. This intense competitive environment increases the survival pressure on app developers. This ultimately affects the integration and utilisation of internal and external innovation resources by developers, as well as the motivation for app innovation [14]. Consequently, the direct effect of negative and positive online reviews on app innovation are influenced by the platform owner's entry. This situation motivates this study to introduce two additional research questions: At the app level, how do app history and app performance moderate the relationship between different online reviews and app innovation? Meanwhile, at the platform owner level, what is the contingent value of the platform owner's entry?

This study makes two principal research contributions. First, given that the impact of different dimensions of online reviews on app innovation remains unclear, we have modelled and tested the relationship between online negative and positive reviews

and app innovation. Second, we explore app history, app performance, and the platform owner's entry as moderating factors for the effects of these online review dimensions on app innovation. Consequently, our study provides insight into the mechanisms underlying the influence of online negative and positive reviews on app innovation.

## 2. Theoretical basis and research hypothesis

### 2.1. Signaling theory

Signaling theory mainly addresses the information asymmetry between two parties in various economic and social contexts. One party possesses information about quality or intention, while the other party is uninformed. Signaling theory illustrates how the informed party communicates this information to the uninformed party through a variety of signals, all of which are intended to achieve favourable outcomes [15]. Signaling theory is closely related to the context of online reviews and the information asymmetry between app users and developers. Notably, language is the primary means of communicating signals among individuals who do not always have equal access to the same information [16].

Signaling theory provides a framework for describing the behaviours of two parties (signal sender and receiver) under conditions of information asymmetry. The signal sender is the party that possesses more information, such as users who provide review information. The receiver is the party that selects useful information from what is transmitted by the sender, such as an app developer. Previous research on signaling theory indicates that users' online reviews, as a type of signal, reduce the information asymmetry of potential users during the product-purchasing process and enhance their decision-making capabilities [17]. In addition, users' online reviews, through the information and emotions conveyed by the signals such as user needs and satisfaction, reduce the information asymmetry of developers during the innovation process, which in turn influences their innovation performance.

Specifically, affective signals and informative signals are the two main attributes embedded in the content of online reviews [18]. The affective attribute expresses emotions or feelings, while the informative attribute provides highly accurate information to reduce the information asymmetry between the signal sender and the signal receiver [19]. This paper argues that two specific types of signals are embedded in online reviews. The first is the affective signal, which is used to convey the signal sender's emotions. The second is the informative signal, which is used to enhance the observability of the signal and reduce the information-acquisition cost for the signal receiver. Online reviews with different features can convey various information or emotions, which ultimately leads to different innovation performances [15].

### 2.2. Negativity bias

Negativity bias indicates that negative emotions and information tend to be more salient, multifaceted, and enduring than positive information [20,21]. Accordingly, research findings indicate that online reviews with more negative ratings are often perceived as more helpful [22]. Moreover, users must invest more cognitive effort and think more carefully when composing online negative reviews [23]. Thus, online negative reviews may offer more diagnostic insights into product quality than their positive counterparts [21,24]. Furthermore, online negative reviews play an important role in the development of online trust [25]. Users typically believe that online negative reviews are more reliable and truthful than positive reviews [9] and the negative information they convey can be considered more helpful for users in avoiding potential risks than that conveyed by online positive reviews [26]. Therefore, online negative reviews are more likely to attract users' attention and carry more weight in their purchase decisions [27].

According to negativity bias, the helpfulness and influence of online negative reviews differ from those of online positive reviews. These differences lead to variations in how developers use online negative and positive reviews to innovate their apps. Online reviews are considered a key resource for developers to continuously refine their apps. The feedback reflecting dynamic user needs, preferences, and app quality enhances developers' innovative ability to grasp market demand and identify innovation opportunities. Additionally, the emotion conveyed by online reviews can affect the developer's

willingness to iterate and innovate the apps [2]. Specifically, online negative reviews provide richer and more diagnostic information than online positive reviews [23]. The signals indicating the need for app updates are often stronger in negative reviews.

Consequently, based on signaling theory and negativity bias, this paper argues that negative and positive reviews convey different information and emotions, which leads to varying effect mechanisms on app innovation.

## 2.3. Research hypotheses

This paper explores the distinct impacts of negative and positive online reviews on app innovation, considering the contingent value of app history, app performance, and platform owner's entry. This investigation is based on the differences in the information and emotional signals conveyed by online negative and positive reviews.

### 2.3.1. Online negative reviews, online positive reviews, and app innovation.

Online negative reviews are evaluations made by users that express dissatisfaction and unfavourable attitudes towards the quality of apps, while online positive reviews are evaluations that convey satisfaction and a supportive attitude towards the quality of apps, typically accompanied by higher overall ratings [28,29]. According to signaling theory and negativity bias, the information and emotions conveyed by online negative and positive reviews differ in both richness and nature. This difference leads to substantial disparities for app developers in aspects such as the acquisition of innovation resources and the reception of emotional signals. Accordingly, this paper posits that the richness of information signals and the strength of emotional signals in online negative reviews can more effectively enhance developers' capabilities and willingness to innovate. The reasons are as follows:

First, online negative reviews convey richer information signals than online positive reviews. This enables app developers to quickly and effectively obtain innovation resources from negative reviews. Online negative reviews provide more feedback regarding user experience with the app than online positive reviews. This helps developers gather user needs effectively, identify innovation opportunities promptly, and enhance their innovation capabilities. The more online negative reviews are given, the more immediate demand information is provided to app developers. This offers more effective insights into functional flaws, performance deficiencies, and the need to add, modify, or remove certain features [7]. The immediate feedback from users on the flaws of the app allows developers to make more targeted adjustments and optimisations, which ultimately enhances their innovation capabilities and shortens the iteration cycle, which in turn promotes rapid innovation of the app [2,30]. Unlike online negative reviews, online positive reviews are affirmations and recognitions of the existing app innovations and contain relatively less information about app flaws or areas for improvement. Consequently, developers find it difficult to extract valuable information from these reviews for app innovation. This makes the role of positive reviews in promoting app innovation weaker than that of negative reviews.

Second, online negative reviews convey a stronger emotional signal for app improvement compared with online positive reviews. Online negative reviews, which reflect negative evaluations, send a stronger signal than online positive reviews that the app functions or services need improvement. This, in turn, enhances developers' willingness to iteratively innovate the app to promptly respond to dynamic user needs and mitigate the negative impact of online negative reviews. Moreover, online negative reviews have a stronger influence on users than online positive reviews. Negative reviews are more likely to influence potential users' purchase decisions [26]. This is attributed to online negative reviews conveying signals of subpar app quality and users' dissatisfaction, which in turn leads to users abandoning the product directly [31], which ultimately threatens the survival of the app. This makes online negative reviews more likely to attract the attention of developers and prompts them to proactively engage in app innovation to meet users' personalised needs. In contrast, online positive reviews convey more favourable signals of app quality to potential users and developers. In addition, the motivational effect of this positive signal on developers' willingness to engage in app innovation is weaker than that of the negative signals [8]. In summary, this paper proposes the following hypothesis:

H1: Online negative reviews promote app innovation more than online positive reviews.

**2.3.2. Moderating role of app performance.** High app performance typically indicates abundant resource reserves and strong innovation capabilities [32], as well as high popularity and a solid user base [33,34]. This paper posits that, compared with online positive reviews, high performance further weakens the positive impact of online negative reviews on app innovation. The reasons are as follows:

First, high performance reduces the advantage of online negative reviews in enhancing the rapid iteration capabilities of the app through the provision of information signals. Furthermore, compared with low-performing apps, high-performing apps have higher resource reserves, which include abundant capital, technology, talent, and market resources at their disposal. These complementary innovation resources enable app developers to overcome resource constraints and gain inspiration and opportunities for app innovation from multiple sources. This creates a certain substitutive effect on the innovation resources obtained from online negative reviews, which in turn reduces the role of these reviews in promoting the innovation capabilities of app developers. Moreover, developers of high-performing apps possess stronger innovation capabilities. Essentially, they have a precise understanding and analysis of user needs. Therefore, developers of high-performing apps may be more inclined to leverage their robust innovation capabilities for app development, which ultimately reduces their reliance on information resources from external user reviews. This weakens the role of online negative reviews in promoting app innovation owing to their information resource advantages. For instance, TikTok is a globally renowned short video sharing app with extremely high performance. It has a huge user base of billions globally and is popular among different age groups and regions. ByteDance, TikTok's parent company, has plenty of capital. This enables heavy investment in server infrastructure, algorithm research, and talent recruitment. Many top engineers and data analysts in the company focus on improving the app's performance and creating new features. They constantly optimize the video-recommendation algorithm to offer more personalized content. Although TikTok gets some negative reviews, like privacy concerns and excessive time- use, these have little impact on its innovation. New video-making effects and interactive features like duet videos were launched because of internal R&D and strategic planning, not mainly negative reviews. With rich resources, TikTok can draw innovation inspiration from various sources, reducing its reliance on negative review information. However, online positive reviews lack this advantage and convey fewer information signals that affect app optimisation and improvement. Therefore, the substitutive effect of high performance on the innovative information resources in online positive reviews is weaker than that on online negative reviews.

Second, high performance reduces the impact of the negative emotional signals conveyed by online negative reviews, which ultimately weakens the stimulating effect of these reviews on developers' willingness to innovate. Compared with low-performing apps, high-performing apps are more popular and have a stronger user base, which provides them with a greater competitive advantage. Potential users tend to choose apps with a large user base and strong network effects. Therefore, although online negative reviews convey a signal of negative evaluation regarding the app quality, high performance reduces the threat these reviews pose to the survival of the app. This, in turn, reduces the attention developers pay to online negative reviews and correspondingly slows their response to user needs highlighted in those reviews, which ultimately weakens the stimulating effect of online negative reviews on developers' willingness to improve the app. Unlike the negative emotional signals conveyed by online negative reviews, online positive reviews provide positive signals about the app to potential users and developers. These positive signals have a limited motivational effect on developers' willingness to innovate [2] and are not weakened by high performance. In contrast, they may be enhanced. In summary, this paper proposes the following hypothesis:

H2: Compared with online positive reviews, weakens the promoting effect of online negative reviews on app innovation.

**2.3.3. Moderating effect of app history.** App history refers to the duration of operation since the release of the app on the platform. An app with a short history is in the early stages of software release, typically in the exploratory or growth phase of development. In contrast, an app with a long history is in the mature phase of development or approaching the decline phase. This paper posits that, compared with online positive reviews, app history more significantly enhances the role of online negative reviews in promoting app innovation. The reasons are as follows:

First, app history improves the positive impact of online negative reviews in enhancing the rapid iteration capabilities of the app by sending information signals. Compared with younger apps, mature apps have a longer survival time, with their development either in the mature phase or nearing the decline phase, where they face slow user growth and increased competition from new products or substitutes. Consequently, developers of mature apps pay more attention to user experience and focus on refined operations. In this context, promptly addressing individual user needs, relying on user participation to obtain more innovative resources, and maintaining continuous updates of the app becomes crucial for developers [35]. Online negative reviews have a more significant advantage in providing information resources compared with online positive reviews. This helps app developers enhance the transformation and utilisation of these resources to promptly capture user needs, improve innovation capabilities, and prevent user churn. In contrast, apps with a short operating history are in the exploration or growth stage. For these apps, the primary goal is to offer users an experience that exceeds their expectations, quickly attract users, and capture market share. However, the innovation of app features or services to deliver an experience beyond users' expectations is challenging to obtain from online reviews provided by users. Instead, this innovation relies more on the developers' professional knowledge and resource innovation [36]. For instance, Clubhouse is a new audio - social app. At its launch, the developers continuously launched new topic rooms, inviting well-known figures, and holding themed audio discussions to attract more users and keep them active. In this process, they relied more on their professional judgment and resource innovation than on user negative reviews for app innovation and improvement. This shows that early-stage apps, due to different development goals and resource use, depend less on negative reviews.

Second, app history amplifies the effect of the negative emotional signals conveyed by online negative reviews. This enhances the stimulating effect of online negative reviews on developers' willingness to innovate. As mentioned earlier, compared with apps with a short operating history, apps in the mature or decline phase face more competition from similar types of apps or substitutes in the market. The negative information more likely results in user churn [37], which makes the emotional signal for improvement stronger for mature apps. Thus, when the app history is long, compared with online positive reviews, strong competitive pressure prompts developers to pay more attention to the feedback from online negative reviews. For example, Alipay has been operating for a long time since its launch and is now in the mature stage. Some complain that the product descriptions in Alipay's wealth management section are unclear and that some complex wealth management products are hard for ordinary users to understand. Alipay's development team values these negative reviews highly. They know that in the mature stage, user experience matters most and refined operations are essential to retain users. So, Alipay has improved its wealth management section in several ways. It optimized the introduction page of products, added plain language explanations and risk warnings, and launched more low risk wealth management products for ordinary users. In summary, this paper proposes the following hypothesis:

H3: Compared with online positive reviews, app history more significantly enhances the positive effect of online negative reviews on app innovation.

**2.3.4. Moderating role of platform owner's entry.** Platform owner's entry refers to the situation in which platform owners imitate existing complementors by entering their market space with similar products to generate profits and maintain control over the ecosystem [13,38]. In the app market, platform owners such as Apple and Google replicate existing apps by developing similar types of apps to participate in platform competition. This turns platform owners into the direct competitors of app developers, which in turn threatens the survival of similar apps and influences developers' innovative behaviour [1]. This paper will investigate the effect of a platform owner's entry on the relationship between online reviews and app innovation. The paper posits that, compared with online positive reviews, the attention overflow and competitive pressure due to platform owner entry can more significantly enhance the role of online negative reviews in promoting app innovation for the following reasons:

First, the attention spillover effect due to the platform owner's entry strengthens information signals, which in turn enhances the mechanism through which app developers use online negative reviews to obtain information resources and

improve innovation capabilities. When the platform owner enters the app market, user attention to similar apps increases. The increased attention helps app developers have greater availability of user feedback [1], including negative and positive reviews. Because of the increased attention about a category, developers may decide to channel innovative efforts and resources toward this category. For example, when Facebook integrated Instagram, a widely favoured photography application, it significantly boosted the user demand for the whole genre of photography apps. The enhancement in users' attention implies that developers possess more information resources from user feedback that can expedite innovation [39]. Moreover, compared with platform owners, developers with inherent resource disadvantages find it particularly essential to obtain additional innovative resources and enhance their innovation capabilities. When the degree of platform owner's entry is high, the increased online negative reviews are more likely to accurately reflect user needs and the shortcomings of the apps. This allows developers to obtain more effective information signals and promptly transform them into innovation capabilities and innovative outputs. However, owing to the relatively ambiguous information conveyed by online positive reviews, the positive moderating effect of platform entry on the relationship between online positive reviews and app iterative innovation is less pronounced.

Second, the competitive pressure due to the platform owner's entry enhances the emotional signals about app quality improvements conveyed by online negative reviews, which more likely stimulates app developers' willingness to innovate. This is attributed to the platform owner's entry leading to more intense competition in the market for similar apps [1]. In a highly competitive environment, compared with online positive reviews, online negative reviews present a greater threat to the survival and development of apps. This compels developers to pay closer attention to online negative reviews and be more inclined to innovate rapidly in response to users' specific needs. Therefore, this paper proposes the following hypothesis:

H4: Compared with online positive reviews, the platform owner's entry more significantly enhances the positive effect of online negative reviews on app innovation Fig 1.

## 3. Research design

### 3.1. Sample selection and data sources

This paper selects the top 500 paid apps in the iOS App Store in China as of December 1, 2018, as the initial sample. The sample period ranges from December 2018 to December 2019. Considering that the study uses one-period lagged

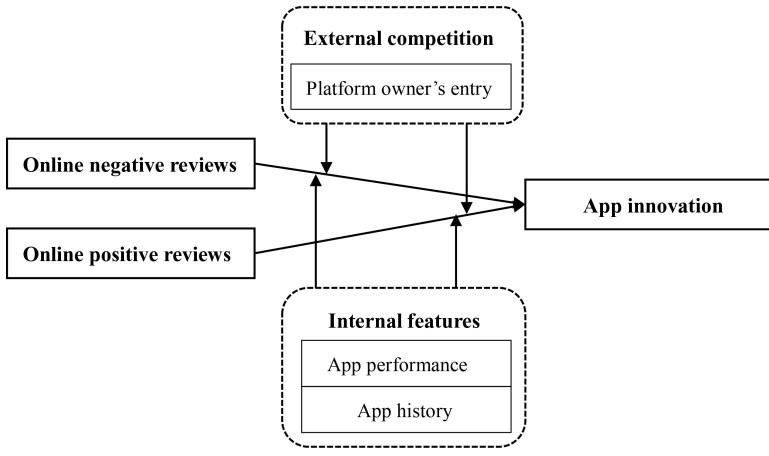

**Fig 1. Shows the research framework of this study.**

data, the data period for analysis is from January 2019 to December 2019. The iOS App Store in China is selected mainly owing to the rapid development of the Chinese app market, where the iOS App Store accounts for > 40% of the country's software application market. The data volume is substantial, and compared with the fragmented Android platform, the data from the iOS App Store is standardised and centralised, which makes it relatively easy to obtain. In addition, selecting the top 500 apps is ideal for this study as this ensures a larger and more stable sample size. Considering that most apps have a short life cycle, those ranked at the top tend to have competitive strength and continuous survival ability [40], which makes them suitable for this research. According to the initial sample, this study excludes apps that are unavailable for information and data retrieval during the research period owing to being delisted or undergoing maintenance. The study also excludes apps published by the iOS platform. A 1%-99% winsorisation treatment is applied to all continuous variables, which in turn results in a final dataset consisting of monthly data for 384 apps across 22 fields. In addition, most game apps generate revenue greater than that of non-game apps. Thus, the data set does not include game apps to avoid the revenue distribution for apps of sample is heavily skewed [41,42].

Furthermore, to test the robustness of our model, we collected the latest available data on the same variables of the initial sample from December 2023 to December 2024. All the data for this study is sourced from Qimai Data (https://www.qimai.cn/), a professional mobile promotion data analysis platform in China. Currently, the platform has more than 7 million apps, with over 12 million app data entries, covering app stores in 155 countries and regions worldwide. The platform provides multi-dimensional data, such as app attributes, real-time operational status, and developer information, which are both abundant and comprehensive. All variable measurement data in this paper are derived from the Qimai Data platform.

### 3.2. Measurements of variables

**3.2.1. Dependent variable.** App innovation. Building on previous research [43], this study uses the number of app updates per month to measure app innovation. The more updates an app receives in a month, the higher and faster the degree of innovation. To account for the lag relative to the independent variable and potential endogeneity issues, this study uses app innovation in period t + 1 as the dependent variable.

**3.2.2. Independent variable.** Online negative reviews/online positive reviews. Previous studies typically classify extreme reviews, such as one-star ratings and five-star ratings, as online negative reviews and online positive reviews, respectively. Building on these studies [44], while avoiding the particularities of extreme reviews and considering the actual context of the app market, this study uses the total number of one-star and two-star reviews added each month to measure online negative reviews. In addition, the study uses the total number of four-star and five-star reviews added each month to measure online positive reviews. To mitigate the effect of large inherent data differences on the results, the obtained data is processed by taking the logarithm.

**3.2.3. Moderator variables.** App performance. Building on previous studies [34,45], this study uses the average monthly ranking on the app bestseller list to measure app performance. In the iOS App Store, the ranking on the app bestseller list is positively correlated with app sales volume. Thus, the higher the ranking, the better the app sales performance and the stronger its profitability.

App history. In this study, we measure app history based on the number of years the app has been in continuous operation in the iOS App Store [34]. The larger the number of app years, the longer the app has been in operation. Platform owner's entry. Building on previous research [13], this study measures platform owner's entry based on the number of platform owners offering apps in the same category that survived throughout the study period. This measure aims to assess the effect of competition from the platform owner on complementary innovation.

**3.2.4. Control variables.** Drawing from previous research, this study controls the following factors that may affect app innovation across three levels: At the app characteristics level, the ranking of download volumes is controlled, measured based on the average monthly ranking of app download volumes; and previous app innovation is also controlled, measured based on the number of updates the app received in the previous month. At the developer level, the

developer's response, which is the number of replies from the app developer to online user reviews in the current month, is controlled and processed logarithmically [43]; the developer's technical experience is measured based on the number of fields in which the developer has developed apps; and the developer's ecosystem experience is measured based on the number of years the developer has gained experience in the iOS platform ecosystem [40]. At the platform level, this study controls the platform recommendation [46], which indicates whether the platform recommends the app in the current month, coded as '1' for 'yes' and '0' for 'no'.

### 3.2.5. Model specification.

This paper uses two-way fixed effects model to test the main and moderating effects. In the following equation, subscript $i$ represents the app and subscript $t$ represents the time.

$$\begin{aligned} Innovation_{i(t+1)} = {} & \beta_0 + \beta_1 Negative\ reviews_{it} + \beta_2\ Positive\ reviews_{it} + \beta_3 App\ performance_{it} + \beta_4 App\ history_{it} \\ & + \beta_5 Platform\ owner's\ entry_{it} + \beta_6 Negative\ reviews_{it} * App\ performance_{it} \\ & + \beta_7 Negative\ reviews_{it} * App\ history_{it} + \beta_8 Negative\ reviews_{it} * Platform\ owner's\ entry_{it} \\ & + \beta_9 Positive\ reviews_{it} * App\ performance_{it} + \beta_{10} Positive\ reviews_{it} * App\ history_{it} \\ & + \beta_{11} Positive\ reviews_{it} * Platform\ owner's\ entry_{it} + Controls_{it} + Month_t + I_i + \omega_{it} \end{aligned}$$

Where $Innovation_{i(t+1)}$ denotes our dependent variable, i.e., app $i$'s innovation at month $t+1$. $Negative\ reviews_{it}$ and $Positive\ reviews_{it}$ are our lagged version of independent variables that the app $i$ has received at time $t$. $App\ performance_{it}$, $App\ history_{it}$ and $Platform\ owner's\ entry_{it}$ are moderating variables. $Controls_{it}$ refers to the control variables mentioned previously. $Month_t$ is time fixed effect. This paper also includes app fixed effect, $I_i$, to control for unobservable time-invariant characteristics of apps, which may explain differences in innovation among apps beyond our explanatory variables. $\omega_{it}$ is the random error term and $\beta$ is a parameter vector.

## 4. Empirical research

### 4.1. Descriptive statistical analysis and correlation coefficient test

According to the results (Table 1), most of the explanatory variables are significantly correlated with the dependent variable, app innovation, which ultimately indicates that the variable selection in this study is reasonable. In addition, the correlation coefficients between all explanatory variables do not exceed the critical value of 0.7 and the variance inflation factor (VIF) for the model is ≤ 2.93, which indicates that no serious multicollinearity issues are present in this study.

### 4.2. Hypotheses testing

The model undergoes a Chow test. The results indicate that the fixed-effects model is superior to the mixed-effects model, with the two-way fixed-effects model being even better. Subsequently, the Hausman test reveals a preference for the fixed-effects model over the random-effects model. Thus, this study employs the two-way fixed-effects model to perform regression analysis on Models 1–6 in Table 2, controlling for both individual fixed effects and temporal trends simultaneously.

In Table 2, model 1 includes only the control variables, while Model 2 incorporates the independent variables, which include the quantity of online negative reviews and online positive reviews, and the moderating variables, which include app performance, app history, and platform owner's entry. The empirical results indicate a significant positive effect of online negative reviews on app innovation (b = 0.053, $p < 0.10$), while no significant effect of online positive reviews on app innovation (b = 0.014, $p > 0.10$) is observed. This suggests that online negative reviews are more likely to promote app innovation than online positive reviews. This finding supports Hypothesis 1.

Drawing from previous research, we incorporate interaction terms sequentially into the main effect model to test each moderating effect separately through Models 3–5. Model 3 adds the interaction terms of online negative reviews with

**Table 1. Descriptive statistical analysis and Pearson correlation coefficient.**

| Variable | Mean | SD | 1 | 2 | 3 | 4 | 5 | 6 | 7 | 8 | 9 | 10 | 11 |
|---|---|---|---|---|---|---|---|---|---|---|---|---|---|
| 1 App innovation$_{t+1}$ | 1.737 | 1.471 | 1 | | | | | | | | | | |
| 2 Online negative reviews | 0.771 | 3.064 | 0.232*** | 1 | | | | | | | | | |
| 3 Online positive reviews | 7.573 | 38.242 | 0.173*** | 0.619*** | 1 | | | | | | | | |
| 4 App performance | 11.173 | 5.001 | 0.178*** | 0.498*** | 0.392*** | 1 | | | | | | | |
| 5 App history | 4.686 | 2.433 | 0.120*** | 0.268*** | 0.161*** | 0.134*** | 1 | | | | | | |
| 6 Platform owner's entry | 4.780 | 5.111 | -0.010 | 0.155*** | 0.012 | -0.052** | -0.018 | 1 | | | | | |
| 7 Download ranks | 5.727 | 6.340 | 0.147*** | 0.636*** | 0.516*** | 0.253*** | 0.222*** | 0.173*** | 1 | | | | |
| 8.Platform recommendations | 0.208 | 0.406 | 0.103*** | 0.212*** | 0.120*** | 0.109*** | 0.049*** | -0.025* | 0.162*** | 1 | | | |
| 9 Developer's response | 1.031 | 1.581 | 0.124*** | 0.311*** | 0.134*** | 0.172*** | 0.043*** | 0.101*** | 0.216*** | 0.230*** | 1 | | |
| 10 Previous innovation | 1.738 | 1.468 | 0.442*** | 0.258*** | 0.195*** | 0.186*** | 0.115*** | -0.009 | 0.138*** | 0.108*** | 0.137*** | 1 | |
| 11 Technical experience | 12.185 | 22.968 | -0.041** | 0.214*** | 0.087*** | 0.095*** | 0.046*** | 0.125*** | 0.175*** | 0.0170 | -0.052** | -0.043** | 1 |
| 12 Ecosystem experience | 6.053 | 2.476 | 0.046*** | 0.259*** | 0.076*** | 0.126*** | 0.668*** | 0.029** | 0.231*** | 0.142*** | 0.030** | 0.040*** | 0.390*** |

Note: *** $p < 0.01$, ** $p < 0.05$, * $p < 0.1$.

app performance and online positive reviews with app performance to the basis of Model 2. The regression results show that the regression coefficient of the interaction term between negative reviews and app performance is negative and significant (b = −0.073, $p < 0.05$), which in turn indicates that app performance significantly negatively moderates the relationship between online negative reviews and app innovation. In contrast, the regression coefficient of the interaction term between online positive reviews and app performance is positive and significant (b = 0.060, $p < 0.05$), which in turn indicates that app performance significantly positively moderates the relationship between online positive reviews and app innovation. These results suggest that, compared with online positive reviews, app performance more effectively weakens the promotional effect of online negative reviews on app innovation. This finding supports Hypothesis 2.

Model 4 includes an interaction term between online negative reviews and app history, as well as between online positive reviews and app history. The regression results show that the regression coefficient of the interaction term between negative reviews and app history is positive and significant (b = 0.100, $p < 0.05$), which in turn indicates that app history significantly and positively moderates the relationship between online negative reviews and app innovation. The regression coefficient of the interaction term between online positive reviews and app history is positive but not significant (b = 0.015, $p > 0.10$), which in turn indicates that app history does not affect the relationship between online positive reviews and app innovation. These results demonstrate that, compared with online positive reviews, app history more effectively enhances the promotional effect of online negative reviews on app innovation. This finding validates Hypothesis 3.

Model 5 includes interaction terms between online negative reviews and platform owner's entry, as well as between online positive reviews and platform owner's entry, to test the moderating effect of platform owner's entry on the relationship between online reviews and app innovation. The results of Model 5 show that the interaction term between online negative reviews and the platform owner's entry has a significant positive effect on app innovation (b = 0.115, $p < 0.05$).

**Table 2. The fixed effects regression results.**

| Variables | App Innovation | | | | | |
|---|---|---|---|---|---|---|
| | Model 1 | Model 2 | Model 3 | Model 4 | Model 5 | Model 6 |
| Online negative reviews | | 0.053* | 0.056* | 0.064** | 0.067** | 0.081** |
| | | (-1.674) | (-1.765) | (-2.008) | (-2.078) | (-2.493) |
| Online positive reviews | | 0.014 | 0.015 | 0.016 | 0.008 | 0.01 |
| | | (-0.906) | (-0.943) | (-0.999) | (-0.518) | (-0.615) |
| Online negative reviews * App performance | | | -0.073** | | | -0.069** |
| | | | (-2.248) | | | (-2.103) |
| Online positive reviews * App performance | | | 0.060** | | | 0.065** |
| | | | (-2.065) | | | (-2.226) |
| Online negative reviews * APP history | | | | 0.100** | | 0.100** |
| | | | | (-2.219) | | (-2.23) |
| Online positive reviews * APP history | | | | 0.015 | | 0.012 |
| | | | | (-0.411) | | (-0.327) |
| Online negative reviews * Platform owner's entry | | | | | 0.115** | 0.115** |
| | | | | | (-2.407) | (-2.387) |
| Online positive reviews * Platform owner's entry | | | | | -0.093*** | -0.096*** |
| | | | | | (-2.608) | (-2.713) |
| App performance | | 0.001** | 0.0001* | 0.001** | 0.001** | 0.001** |
| | | (-2.447) | (-2.046) | (-2.415) | (-2.335) | (-2.030) |
| App history | | 0.156 | 0.164 | 0.161 | 0.159 | 0.173 |
| | | (-0.848) | (-0.893) | (-0.877) | (-0.864) | (-0.941) |
| Platform owner's entry | | -0.008 | -0.009 | -0.009 | -0.018 | -0.02 |
| | | (-0.592) | (-0.668) | (-0.679) | (-1.262) | (-1.389) |
| Download ranks | 0.0001** | 0.0001 | 0.0001 | 0.001 | 0.001 | 0.001 |
| | (-2.09) | (-1.47) | (-1.483) | (-1.606) | (-1.472) | (-1.619) |
| Platform recommendation | 0.031 | 0.025 | 0.027 | 0.024 | 0.028 | 0.029 |
| | (-0.407) | (-0.327) | (-0.352) | (-0.309) | (-0.362) | (-0.377) |
| Developer's response | 0.018 | -0.005 | -0.003 | -0.002 | -0.004 | -0.000 |
| | (-0.693) | (-0.175) | (-0.115) | (-0.070) | (-0.162) | (-0.003) |
| Previous innovations | -0.054*** | -0.062*** | -0.062*** | -0.063*** | -0.063*** | -0.066*** |
| | (-3.488) | (-3.939) | (-3.999) | (-4.027) | (-4.053) | (-4.197) |
| Technical experience | -0.013 | -0.013 | -0.014 | -0.015 | -0.019 | -0.021 |
| | (-0.868) | (-0.913) | (-0.917) | (-1.025) | (-1.292) | (-1.422) |
| Ecosystem experience | 0.063 | 0.003 | -0.000 | 0.002 | 0.005 | -0.001 |
| | (-0.442) | (-0.020) | (-0.001) | (-0.013) | (-0.032) | (-0.005) |
| Time | Yes | Yes | Yes | Yes | Yes | Yes |
| Constant | -2.735 | -7.409 | -7.581 | -7.658 | -7.597 | -7.974 |
| | (-0.264) | (-0.642) | (-0.658) | (-0.664) | (-0.659) | (-0.692) |
| R2 | 0.502 | 0.505 | 0.506 | 0.506 | 0.506 | 0.508 |
| N | 4582 | 4564 | 4564 | 4564 | 4564 | 4564 |

Note: *** $p<0.01$, ** $p<0.05$, * $p<0.1$.

The interaction term between online positive reviews and the platform owner's entry has a significant negative effect on app innovation (b = −0.093, $p < 0.01$). Therefore, compared with online positive reviews, the platform owner's entry more effectively enhances the positive effect of online negative reviews on app innovation. This finding supports Hypothesis 4. Model 6 includes interaction terms for the three moderating variables with both online negative and positive reviews. The regression results are consistent with those from Models 2–5, which ultimately indicates the robustness of the results for Hypotheses 2–4 and further supports these hypotheses in this paper.

### 4.3. Endogeneity and robustness tests

**4.3.1. Endogeneity.** Potential endogeneity problems may stem from reverse causality, omitted variables, and the like [42]. For example, there may also be a reverse causal relationship between app innovation and online reviews: a high degree of app innovation prompts the users to provide more feedback about the quality of services conveyed by online reviews [4]. In addition, omitting variables such as the development trend of industry technology, which simultaneously affect both the app innovation and online reviews, may lead to biased estimation results [47]. To address the potential endogeneity issues, this paper uses the following methods. First, regarding the potential bidirectional causality between the independent and dependent variables, the app innovation indicators measured in this paper are all lagged by one period. The app innovation indicators for period t + 1 will not affect the number of online negative and positive reviews in period t, thus effectively avoiding the issue of mutual causality. Second, endogeneity issues may also arise from omitted variable factors. This study uses panel data to mitigate the problem of omitted variables (individual differences among apps) to some extent.

Furthermore, we also address potential endogeneity issue by using the 2SLS method that is often used to deal with various endogeneity problems [48]. Through identifying suitable instrumental variables and performing corresponding estimations across the two-stage procedure, the 2SLS method can effectively tackle intricate endogeneity structures. The selection of instrumental variables necessitates their correlation with the endogenous explanatory variables while maintaining no correlation with the error terms. Thus, drawing on related research [43,49], the average number of new online negative reviews and positive reviews per month for similar category apps, excluding the current app, are selected as instrumental variables. The underlying logic is as follows: on the one hand, there exists a strong positive correlation between the average number of online reviews of apps in the same category and those of the current app. Specifically, the higher the average number of online reviews of apps in the same category, the more likely it is that the current app will have a higher number of online reviews. On the other hand, the average number of online reviews of apps in the same category clearly does not directly influence the current app's innovation, thus satisfying the correlation and exogeneity assumption of instrumental variables [50].

Model 1 (Table 3) shows the results of the 2SLS estimation, where the coefficient of online negative reviews is significantly positive, while the coefficient of online positive reviews is not significant. This is completely consistent with the main test results in Model 6 (Table 2). The other regression results are also generally consistent with the previous test results. This indicates that after controlling for endogeneity, the conclusions of this study remain robust.

**4.3.2. Robustness tests.** The robustness tests in this study include two aspects: (1) Variable substitution: Drawing from previous research, extreme reviews are used, where the number of one-star reviews measures online negative reviews and the number of five-star reviews measures online positive reviews. In Table 3, the regression results in Model 2 show that the regression coefficient of online negative reviews and app history is positive and significant (b = 0.078, p < *0.05*), while that of online positive reviews and app history is positive but not significant (b = 0.007, p > 0.10), which are generally consistent with the main results of the previous study and show no substantial changes, demonstrating the robustness of the research findings. In addition, this study substitutes the dependent variable. Specifically, this study re-measures app innovation based on the number of specific content updates across different versions of the app. The regression results in Model 3 indicate that online

**Table 3. Robustness and endogeneity tests.**

| Variable | 2SLS | Changing IV | Changing DV | Subsample | Sample of 2024 |
|---|---|---|---|---|---|
| | Model 1 | Model 2 | Model 3 | Model 4 | Model 5 |
| Online negative reviews | 0.692** | 0.078** | 0.248* | 0.146*** | 0.087** |
| | (-2.228) | (-2.51) | (-1.939) | (-3.319) | (2.316) |
| Online positive reviews | 0.175 | 0.007 | 0.101 | 0.031 | 0.001 |
| | (-0.241) | (-0.473) | (-1.566) | (-1.363) | (0.035) |
| Online negative reviews * App performance | -0.088** | -0.067** | -0.082 | -0.104* | -0.134* |
| | (-2.159) | (-2.049) | -0.642 | (-1.669) | (-1.719) |
| Online positive reviews * App performance | 0.091*** | 0.063** | 0.251** | 0.069 | 0.147* |
| | (-2.748) | (-2.185) | (-2.213) | (-1.313) | (1.655) |
| Online negative reviews * App history | 0.220*** | 0.117*** | 0.420** | 0.173*** | 0.224*** |
| | (-2.686) | (-2.687) | (-2.391) | (-2.963) | (2.964) |
| Online positive reviews * App history | 0.061 | 0.008 | 0.085 | -0.025 | -0.040 |
| | (-0.216) | (-0.215) | (-0.588) | (-0.523) | (-0.535) |
| Online negative reviews *Platform entry | 0.273*** | 0.103** | 0.661*** | 0.129** | 0.065 |
| | (-2.743) | (-2.176) | (-3.498) | (-2.03) | (0.828) |
| Online positive reviews *Platform entry | -0.056 | -0.098*** | -0.268* | -0.102** | -0.111* |
| | (-0.226) | (-2.814) | (-1.924) | (-2.369) | (-1.707) |
| App performance | -0.001 | 0.001** | 0.001** | 0.001 | -0.001 |
| | (-0.593) | (-2.058) | (-2.127) | (-0.371) | (-1.566) |
| App history | 0.108 | 0.176 | 0.147 | 0.002 | 0.008 |
| | (-0.692) | (-0.962) | (-0.205) | (-0.008) | (0.103) |
| Platform owner's entry | -0.031* | -0.02 | -0.032 | -0.012 | 0.013 |
| | (-1.788) | (-1.379) | (-0.556) | (-0.636) | (0.532) |
| Download ranks | -0.001 | 0.001* | 0.001 | 0.001 | -0.035 |
| | (-0.706) | (-1.693) | (-0.226) | (-1.102) | (-0.749) |
| Platform recommendation | -0.015 | 0.028 | 0.231 | 0.076 | 0.019 |
| | (-0.132) | (-0.368) | (-0.763) | (-0.778) | (0.151) |
| Developer's response | -0.144 | 0.003 | 0.067 | 0.009 | -1.912 |
| | (-1.037) | (-0.105) | (-0.613) | (-0.266) | (-2.051) |
| Previous innovations | -0.129* | -0.066*** | -0.447*** | -0.067** | -0.004 |
| | (-1.933) | (-4.212) | (-5.609) | (-3.475) | (-0.417) |
| Technical experience | -0.018 | -0.022 | -0.069 | -0.028 | -0.018 |
| | (-0.475) | (-1.450) | (-1.183) | (-1.630) | (-0.475) |
| Ecosystem experience | -0.056 | -0.004 | 0.019 | 0.076 | -0.021 |
| | (-0.350) | (-0.024) | (-0.030) | (-0.377) | (-0.262) |
| Time | Yes | Yes | Yes | Yes | Yes |
| Constant | -8.113 | -7.95 | -5.905 | -4.223 | -31.337 |
| | (-0.70) | (-0.691) | (-0.131) | (-0.261) | (-1.440) |
| R2 | 0.084 | 0.508 | 0.445 | 0.507 | 0.405 |
| N | 4564 | 4568 | 4564 | 2935 | 2810 |

Note: *** p<0.01, ** p<0.05, * p<0.1.

negative reviews enhance app innovation (b = 0.248, p < 0.1), while online positive reviews have no significant effect on app innovation (b = 0.101, p > 0.10). the main results are generally consistent with the previous study. This further demonstrates the robustness of the research findings. (2) Sub-sample regression: The top 300 ranked apps on the bestseller list are selected as an independent sub-sample to establish regression models, considering that these apps have stronger survival capabilities and relatively more user feedback. The regression results in Model 4 demonstrate that online negative reviews have a positive relationship with app innovation (b = 0.146, p < 0.1), while online positive reviews have no significant effect on app innovation (b = 0.031, p > 0.10). The main conclusions remain unchanged.

In addition, this study re-collected the latest available data on the same variables of the initial sample (the top 300 ranked apps) from December 2023 to December 2024 to further address concerns about data timeliness. The re-collected data was sourced from the Apple App Store in the Chinese market. In 2023–2024, Apple demonstrated a stronger inclination towards launching its own applications, and users' attention decreased substantially. Thus, compared with the situation in 2018–2019, the market environment in which apps operated in 2023–2024 experienced more intense competition. Continuous innovation has become increasingly crucial for developers, with the two sets of data from different periods being similar.

Table 4 presents the descriptive statistics of the latest data. Compared with the statistics in Table 1, the mean of app innovation (Table 4) increases by 0.427 and the standard deviation increases by 0.273. This result indicates a higher level of app innovation and a wider innovation gap among apps. For user reviews, the mean of online negative reviews increases by 0.502, while its standard deviation decreases by 0.128. Conversely, the mean of online positive reviews decreases by 1.713, with its standard deviation decreasing by 5.293. These changes indicate that users' demands for apps have become more sophisticated, which in turn reflects greater diversity and personalization. For the moderated variables, the mean of app history increases by 0.073, accompanied by a slight decrease in standard deviation. The mean of app performance decreases by 1.921, whereas the standard deviation increases by 2.759, which ultimately reflects intensified competition among apps and a growing performance disparity. Additionally, the mean of platform owner's entry increases by 0.45, with the standard deviation increasing by 0.299, which in turn indicates that platform owners contribute to increased market competition. Overall, while the latest data demonstrates changes compared with 2018–2019 data, the differences between the two datasets are not significant.

In Table 3, the regression results in Model 5 show that the empirical results indicate a significant positive effect of online negative reviews on app innovation (b = 0.087, p < 0.05), while no significant effect of online positive reviews on app innovation (b = 0.001, p > 0.10) is observed. The main results remained unchanged. Specifically, this study uses the two-way fixed effects model to test the hypotheses, effectively controlling for the influence of time factors in the regression analysis (Table 2). Thus, our findings are robust and not subject to one specific time.

**Table 4. Descriptive statistical analysis of the latest data.**

| Number | Variable | Mean | SD |
|---|---|---|---|
| 1 | App innovation | 2.164 | 1.744 |
| 2 | Online negative reviews | 1.273 | 2.918 |
| 3 | Online positive reviews | 5.860 | 32.949 |
| 4 | App performance | 9.252 | 7.760 |
| 5 | App history | 4.759 | 2.402 |
| 6 | Platform owner's entry | 5.230 | 5.410 |

## 5. Conclusions and discussion

### 5.1. General conclusions

This study uses monthly balanced panel data from apps to empirically analyse the effect of online negative and positive reviews on app innovation. The results indicate that online negative reviews significantly promote app innovation, while online positive reviews do not have a significant effect on app innovation. Therefore, this paper argues that online negative reviews are more effective in promoting app innovation than online positive reviews. Additionally, compared with online positive reviews, app performance more effectively weakens the promoting effect of online negative reviews on app innovation. Moreover, both app history and platform owner's entry play a positive moderating role in the impact of online negative reviews on app innovation, while no positive moderating effect is observed in the impact of online positive reviews on app innovation. This result indicates that under different contingencies, the differences in the impact of online negative and positive reviews on app innovation will be further amplified.

### 5.2. Theoretical contributions

The findings of this study extend the research on online reviews and app innovation, with the following theoretical contributions.

(1)  This study examines the role of online reviews in app innovation from the perspectives of signaling theory and negativity bias theory. The study highlights the differences in signals between online negative and positive reviews and verifies the various strategic values of these reviews for app innovation. The prevalence of online reviews has garnered substantial attention in research on this topic. However, most of these studies are based on early word-of-mouth research, which explores the impact of online reviews on users' purchasing behaviour and sales volume [51,52]. Recently, few studies have started focusing on the value of online reviews in the innovation process. These studies emphasise that the feedback shared by users through online reviews serves as an important resource for product innovation. Moreover, previous empirical studies have verified that users' online reviews, such as the number of reviews and review ratings, can significantly promote app innovation [3]. However, unlike those studies, this research considers the internal characteristics of online reviews by further dividing them into two dimensions—online negative and positive reviews—based on the nature of the reviews. This study empirically tests the different impacts of online negative and positive reviews on app innovation, which in turn supplements the empirical research on the relationship between online reviews and innovation.

(2) This study elucidates the contingent value of app characteristics and platform owner's entry in the relationship between online reviews and app innovation. Current research on online reviews and app innovation is still in its infancy and no studies have yet deeply explored the contingency mechanisms through which online reviews influence app innovation. Given the characteristics of the platform ecosystem, the outcomes of app innovation are inevitably influenced by multiple entities such as the platform, users, and the app itself. Accordingly, this study, from the user's perspective, examines the disparate effects of negative and positive online reviews on app innovation. This examination scrutinises the contingent value of three pivotal factors: platform owner's entry, app performance, and app history. The results reveal that, compared with positive reviews, app performance more effectively weakens the positive effect of online negative reviews on app innovation, while app history and platform owner's entry both more effectively enhance the positive effect of online negative reviews on app innovation. These findings provide a comprehensive understanding of the differentiated mechanisms through which online negative and positive reviews affect app innovation and enrich the discussion on the influence of online reviews on app innovation.

### 5.3. Practical implications

The findings of this study have practical implications for app developers and platform owners. From the perspective of developers, to encourage users to provide effective reviews and drive app innovation, the following measures can be implemented: First, as demonstrated in this paper, online negative reviews have a more substantial promotional effect on app innovation compared with online positive reviews. Therefore, developers should refrain from deliberately encouraging users to submit false reviews or assign inflated ratings. Second, given that reviews with stronger information signals more effectively drive app innovation, developers can introduce incentive mechanisms. For example, they can offer vouchers, loyalty points, or monetary incentives to encourage users to contribute authentic and valuable review content. Third, developers can establish guidelines for writing online reviews. These guidelines should provide users with helpful suggestions on how to write reviews and encourage them to share more detailed product usage information and insights about their user experience. Fourth, developers can leverage artificial intelligence and text-mining tools to conduct semantic analysis. This approach can effectively optimise the filtering mechanism for useful reviews, enhance the efficiency of information extraction, and ultimately contribute to app innovation.

In addition, for platform owners, to effectively stimulate developers' innovation, the following measures can be implemented: First, platform owners should integrate their resources, appropriately participate in platform competition, motivate other developers to innovate positively, and maintain the sustainable development of the platform ecosystem. Second, platform owners should establish an accurate and fair review ecosystem to minimise the generation of false, high-rated yet useless reviews.

### 5.4. Limitations and future research directions

This study has some limitations that require further improvement in future research. First, the sample in this study is limited to iOS apps in the Chinese region. Considering that statistical methods and app categories vary greatly among domestic mobile brand app markets, the Android app market is not included. Essentially, the difference in openness between Android and iOS app markets may lead to different innovation levels among developers on these platforms. Future research should include samples from other countries to explore how online comments, especially positive and negative reviews, affect app innovation in both the iOS App Store and Google Play. Second, the sample in this study does not incorporate games. Nevertheless, games are characterised by frequent updates, a substantial user base, and a significant volume of comments. Future research should focus on apps in the gaming category to investigate how different types of online reviews affect app innovation.

## Author contributions

**Conceptualization:** Lin Sun.

**Data curation:** Yuting He.

**Funding acquisition:** Lin Sun, Feng Fu.

**Investigation:** Feng Fu.

**Methodology:** Yuting He, Feng Fu.

**Writing – original draft:** Lin Sun, Feng Fu.

**Writing – review & editing:** Lin Sun, Yuting He, Feng Fu.

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
