## [Decision Letter · Decision Letter 0]

7 Jan 2025

PONE-D-24-42585The Impact of Online reviews on App Innovation and Its contingent EffectPLOS ONE

Dear Dr. Fu,

Thank you for submitting your manuscript to PLOS ONE. After careful consideration, we feel that it has merit but does not fully meet PLOS ONE’s publication criteria as it currently stands. Therefore, we invite you to submit a revised version of the manuscript that addresses the points raised during the review process.

**The manuscript explores an innovative topic, examining the differential effects of online reviews on app innovation, yet requires major revisions to meet publication standards. The theoretical framework should integrate complementary perspectives, such as signaling theory or absorptive capacity, to enhance the analysis. The dataset's scope requires a stronger justification or broader dataset should be included to address applicability concerns Figures may be redrawn or cited with proper permissions, and the conclusion should align more clearly with the abstract and title, reflecting key findings. References must be updated to include recent and relevant studies only.**

We look forward to receiving your revised manuscript.

Kind regards,

Ayesha Maqbool, PhD

Academic Editor

PLOS ONE

**Journal Requirements:**

Reviewers' comments:

Reviewer's Responses to Questions

**Comments to the Author**

1. Is the manuscript technically sound, and do the data support the conclusions?

Reviewer #1: Partly

Reviewer #2: Partly

Reviewer #3: Yes

2. Has the statistical analysis been performed appropriately and rigorously? 

Reviewer #1: Yes

Reviewer #2: Yes

Reviewer #3: I Don't Know

3. Have the authors made all data underlying the findings in their manuscript fully available?

Reviewer #1: Yes

Reviewer #2: Yes

Reviewer #3: Yes

4. Is the manuscript presented in an intelligible fashion and written in standard English?

Reviewer #1: No

Reviewer #2: Yes

Reviewer #3: Yes

5. Review Comments to the Author

**Reviewer #1:**  The manuscript addresses an interesting and novel topic, exploring the differential effects of online negative and positive reviews on app innovation. While the study's objective is innovative, the execution falls short, and the current version does not meet the standards for publication. Significant improvements are required in the following areas to enhance its quality and contribution.

The language and clarity require substantial improvement, with numerous grammatical errors and inconsistent phrasing detracting from the paper’s readability. A professional edit is essential to meet international academic standards.

The theoretical framework focuses primarily on negativity bias theory, which is valid and relevant. However, incorporating complementary perspectives, such as signaling theory or absorptive capacity, could enrich the discussion and provide a broader understanding of the mechanisms at play.

The restricted dataset and sample selection, focused only on iOS apps in China and excluding certain app categories like gaming, limits the generalizability of the findings. While this choice may reflect specific research objectives, the authors should provide a clear justification for this scope to address potential concerns about applicability.

The practical implications lack actionable guidance for app developers. Providing specific strategies for leveraging online reviews in innovation would enhance the study’s relevance and impact.

**Reviewer #2:**  Comments to “The Impact of Online reviews on App Innovation and Its contingent Effect”

1. Line 3-5 of “Introduction”, please add the data source.

2. Be careful to proof-read the whole manuscript as it looks like there are quite some grammatical errors. For example:

a) “Insert Figure 1 about here” might improve if replaced by “Insert Figure 1 here”;

b) “3.2.4 Control Variable” would be better if replaced by “3.2.4 Control Variables”;

c) “Due to use one period lagged data, the data period used in this study is from January 2019 to December 2019.” This sentence is poorly written and need to polish.

3. The sample period of your study ranges from 2018 to 2019, internet and mobile devices develop rapidly, your data is too old for readers to know the change currently. Maybe you should update your database and focus on the last 2 years.

4. I did not find the regression equation in your study, please add equations that your study related to.

5. Each hypothesis you accept, or reject should be explained in detail. The reasons why the hypothesis is accepted or rejected this study are not adequate now.

**Reviewer #3:**  - The article is well-organized, with distinct sections for endogeneity, robustness tests, results, theoretical contributions, practical implications, and limitations. This makes it easy for readers to follow the flow of the research.

- The use of lagged variables, instrumental variables, and sub-sample regressions demonstrates a strong commitment to addressing endogeneity and ensuring robustness.

-The discussion on negativity bias theory and the contingent value of app characteristics and platform entry is insightful and adds depth to the understanding of online reviews and app innovation.

Suggestions for Improvement:

- While the methodological rigor is commendable, the explanations, especially for the 2SLS and instrumental variable choice, could benefit from simplification for readers less familiar with econometrics. For instance, the rationale for using the average number of reviews from similar apps as instruments could be expanded to clarify their validity further.

-The terms like "mutual causality" and "omitted variables" could be briefly explained or referenced for readers unfamiliar with econometric challenges.

-Phrases such as "Insert Table 3 about here" disrupt the narrative. Consider integrating the description of Table 3's key results into the text, providing readers with immediate insights without needing the table.

-The section mentions that app history and platform entry positively moderate the impact of negative reviews, while app performance weakens it. This could be elaborated with examples or hypothetical scenarios to make these findings more tangible.

To sum up

The article offers a significant contribution to understanding the relationship between online reviews and app innovation. To enhance its impact, consider refining the clarity of the methodological sections and adding depth to the discussion of results and practical implications. Simplifying econometric terminology and integrating examples could make the research more accessible to a broader audience.

6. PLOS authors have the option to publish the peer review history of their article (what does this mean? ). If published, this will include your full peer review and any attached files.

**Do you want your identity to be public for this peer review?** For information about this choice, including consent withdrawal, please see our Privacy Policy .

Reviewer #1: No

Reviewer #2: No

Reviewer #3: No

---

## [Author Response · Author response to Decision Letter 1]

24 Feb 2025

Response to the AE

Dear Dr. Ayesha Maqbool

Thank you very much for the referee report and the valuable comments on our manuscript, “The Impact of Online reviews on App Innovation and Its contingent Effect” (Manuscript ID: PONE-D-24-42585). We are extremely grateful to your decision on the three reviews.

We have carefully studied the comments from the review team and tried our best to respond to the issues raised by the review team. In this response letter, we first describe the major changes we have made to the manuscript in response to your comments, and we then address the individual issues raised by the three reviewers.

Comments are in italics and our responses are in normal font:

1. The theoretical framework should integrate complementary perspectives, such as signaling theory or absorptive capacity, to enhance the analysis.

Response: We have added signaling theory and revised the section on research hypotheses to strengthen the reasons behind the formulation of the hypotheses (see, the subsection “2.1 Signaling Theory” on Page 2- Page 3�the lines marked in red on Page 4�), and provided practical cases to make the hypotheses more persuasive (see, the lines 4-15 of paragraph 2 on Page 4; the lines 14-24 of paragraph 2 on Page 5; the lines 12-22 of paragraph 2 on Page 6).

2. The dataset's scope requires a stronger justification or broader dataset should be included to address applicability concerns Figures may be redrawn or cited with proper permissions.

Response: In this revision, we further provided the reasons and reference basis for the sample selection (see, the lines marked in red of subsection “3.1 Sample Selection and Data Sources” on Page 9), and re-collected the data from December 2023 to 2024 to verify the main conclusions. The regression results show that the main results remained unchanged. suggesting that our findings are robust and not subject to one specific time (see, the lines 19-25 of paragraph 2 in subsection “4.3.2 Robustness Tests” on Page 14; Model 5 in Table 3 on Page 14). In addition, we have redrawn charts and include them as part of your main manuscript.

3. The conclusion should align more clearly with the abstract and title, reflecting key findings.

Response: In this revision, we have revised the title to “Impact of online negative and positive reviews on App Innovation” and refined the abstract to highlight the key conclusions (see, the lines marked in red of subsection “Abstract” on Page 1).

4. References must be updated to include recent and relevant studies only.

Response: In this revision, we add recent and relevant studies, such as reference [3], [10] and [38] in subsection “References”.

Below we provide more detailed responses to the comments by the two reviewers

Thanks again!

Response to Reviewer 1

Thank you very much for your great comments and suggestions on our manuscript. We are so encouraged that you think the subject and findings are interesting and novel. We have carefully studied the comments from you and tried our best to respond to the issues raised by the review team.

Below we provide a point-to-point response to your comments:

1. The manuscript addresses an interesting and novel topic, exploring the differential effects of online negative and positive reviews on app innovation. While the study's objective is innovative, the execution falls short, and the current version does not meet the standards for publication. Significant improvements are required in the following areas to enhance its quality and contribution.

The language and clarity require substantial improvement, with numerous grammatical errors and inconsistent phrasing detracting from the paper’s readability. A professional edit is essential to meet international academic standards.

Response: Thank you for the constructive suggestion! Following your suggestion, this paper has been checked and corrected for proper English language, grammar, punctuation, spelling, and overall style by the highly-qualified, native English speaking editors at Native English Editing. The statement of edit is attached to Appendix I.

2. The theoretical framework focuses primarily on negativity bias theory, which is valid and relevant. However, incorporating complementary perspectives, such as signaling theory or absorptive capacity, could enrich the discussion and provide a broader understanding of the mechanisms at play.

Response: This is a really great suggestion and thank you very much! In this revision, following your suggestion, we have added signaling theory in the section “2 Theoretical Basis and Research Hypothesis” (see, the subsection “2.1 Signaling Theory” on Page 2- Page 3) and gave a detailed explanations on the mechanisms of online negative reviews on app innovation in the subsection “2.3 Research Hypotheses” (see, the lines marked in red on Page 4).

3. The restricted dataset and sample selection, focused only on iOS apps in China and excluding certain app categories like gaming, limits the generalizability of the findings. While this choice may reflect specific research objectives, the authors should provide a clear justification for this scope to address potential concerns about applicability.

Response: Thanks for pointing this out! Following your suggestion, we further provided the reasons and reference basis for the sample selection (see, the lines marked in red of subsection “3.1 Sample Selection and Data Sources” on Page 9), and proposed that future research could separately examine the influence mechanism of online negative reviews on app innovation within the gaming industry (see, the subsection “5.4 Limitations and Future Research Directions” on Page 17).

4. The practical implications lack actionable guidance for app developers. Providing specific strategies for leveraging online reviews in innovation would enhance the study’s relevance and impact.

Response: Thanks for pointing this out! In this revision, we have rewritten the practical significance to provide actionable guidance for app developers. (see, the subsection “5.3 Practical Implications” on Page 17).

Response to Reviewer 2

Thank you very much for your valuable comments and very constructive suggestions in our paper. We have tried our best to revise the paper to incorporate all your suggestions and address all your concerns. Below we detail how we changed the paper in response to your comments and suggestions.

1. Line 3-5 of “Introduction”, please add the data source.

Response: Thanks for pointing this out! In this revision, we have added the data source. (see, the 3-5 lines in section “1. Introduction” on Page 1).

2. Be careful to proof-read the whole manuscript as it looks like there are quite some grammatical errors. For example:

a) “Insert Figure 1 about here” might improve if replaced by “Insert Figure 1 here”;

b) “3.2.4 Control Variable” would be better if replaced by “3.2.4 Control Variables”;

c) “Due to use one period lagged data, the data period used in this study is from January 2019 to December 2019.” This sentence is poorly written and need to polish.

Response: Thanks for pointing this out! In this revision, the above grammatical errors have been revised (see, the subsection “3.2.4 Control Variables” on Page 9 and the 3-4 lines in subsection “3.1 Sample Selection and Data Sources” on Page 8). Moreover, this paper has been checked and corrected for proper English language, grammar, punctuation, spelling, and overall style by the highly-qualified, native English speaking editors at Native English Editing. The statement of edit is attached to Appendix I.

3. The sample period of your study ranges from 2018 to 2019, internet and mobile devices develop rapidly, your data is too old for readers to know the change currently. Maybe you should update your database and focus on the last 2 years.

Response: Thanks for pointing this out! Following your suggestions, we re-collected the data from December 2023 to 2024 to verify the main conclusions. The regression results in Model 5 (Table 3) show that the main results remained unchanged. suggesting that our findings are robust and not subject to one specific time (see, the lines 19-25 of paragraph 2 in subsection “4.3.2 Robustness Tests” on Page 14; Model 5 in Table 3 on Page 14).

4. I did not find the regression equation in your study, please add equations that your study related to.

Response: Thanks for pointing this out! Following your suggestions, we have added a new subsection “3.2.5 Model Specification” to explain our research equation (see, the subsection “3.2.5 Model Specification” on Page 10).

5. Each hypothesis you accept, or reject should be explained in detail. The reasons why the hypothesis is accepted or rejected this study are not adequate now.

Response: Thanks for pointing this out! Following your suggestions, we have added signaling theory and revised the section on research hypotheses to strengthen the reasons behind the formulation of the hypotheses (see, the subsection “2.1 Signaling Theory” on Page 2- Page 3�the lines marked in red on Page 4�), and provided practical cases to make the hypotheses more persuasive (see, the lines 4-15 of paragraph 2 on Page 4; the lines 14-24 of paragraph 2 on Page 5; the lines 12-22 of paragraph 2 on Page 6). According to the results of the regression analysis in table2, all hypotheses have been accepted (see, the table 2 on Page 12-13).

Response to Reviewer 3

Thank you very much for your great comments and suggestions on our manuscript. We are so encouraged that you think the article is well-organized. We have carefully studied the comments from you and tried our best to respond to the issues raised by the review team.

Below we provide a point-to-point response to your comments:

1. - While the methodological rigor is commendable, the explanations, especially for the 2SLS and instrumental variable choice, could benefit from simplification for readers less familiar with econometrics. For instance, the rationale for using the average number of reviews from similar apps as instruments could be expanded to clarify their validity further.

Response: Thanks for pointing this out! Following your suggestions, in this revision, we further explain the 2SLS method (see, 1-6 lines of paragraph 2 in the subsection “4.3.1 Endogeneity” on Page 13) and strengthen the reasons for selecting the instrumental variables (see, 6-15 lines of paragraph 2 in the subsection “4.3.1 Endogeneity” on Page 13-14).

2. -The terms like "mutual causality" and "omitted variables" could be briefly explained or referenced for readers unfamiliar with econometric challenges.

Response: Thanks for pointing this out! Following your suggestions, we explain the two terms "mutual causality" and "omitted variables", and provided the related references (see, paragraph 1 in the subsection “4.3.1 Endogeneity” on Page 13).

3. - Phrases such as "Insert Table 3 about here" disrupt the narrative. Consider integrating the description of Table 3's key results into the text, providing readers with immediate insights without needing the table.

Response: Thank you for pointing it out! Following your suggestions, we delete "Insert Table 3 about here" and add the description of Table 3's key results into the text to make it convenient for readers to read (see, paragraph 2 in the subsection “4.3.2 Robustness Tests” on Page 14).

4. - The section mentions that app history and platform entry positively moderate the impact of negative reviews, while app performance weakens it. This could be elaborated with examples or hypothetical scenarios to make these findings more tangible.

Response: Thank you very much for your valuable comments on our manuscript! Following your suggestions, we add practical examples to make the moderating effect more persuasive and tangible (see, the lines 4-15 of paragraph 2 on Page 4; the lines 14-24 of paragraph 2 on Page 5; the lines 12-22 of paragraph 2 on Page 6).

Thanks again!

Appendix I.

---

## [Decision Letter · Decision Letter 1]

17 Mar 2025

PONE-D-24-42585R1Impact of online negative and positive reviews on App InnovationPLOS ONE

Dear Dr. Fu,

Thank you for submitting your manuscript to PLOS ONE. After careful consideration, we feel that it has merit but does not fully meet PLOS ONE’s publication criteria as it currently stands. Therefore, we invite you to submit a revised version of the manuscript that addresses the points raised during the review process.

Thank you for your revisions. Your manuscript has improved significantly in clarity, methodological precision, and theoretical grounding. However, a few minor revisions are still required to ensure consistency and completeness. Specifically, there are inconsistencies in the dataset description . Additionally, a stronger justification for contributions should be provided, explaining how it ensures timeliness and relevance. Addressing these points will further strengthen the manuscript's coherence and impact.

We look forward to receiving your revised manuscript.

Kind regards,

Ayesha Maqbool, PhD

Academic Editor

PLOS ONE

Journal Requirements:

Reviewers' comments:

Reviewer's Responses to Questions

**Comments to the Author**

1. If the authors have adequately addressed your comments raised in a previous round of review and you feel that this manuscript is now acceptable for publication, you may indicate that here to bypass the “Comments to the Author” section, enter your conflict of interest statement in the “Confidential to Editor” section, and submit your "Accept" recommendation.

Reviewer #1: (No Response)

Reviewer #2: All comments have been addressed

2. Is the manuscript technically sound, and do the data support the conclusions?

Reviewer #1: Partly

Reviewer #2: Yes

3. Has the statistical analysis been performed appropriately and rigorously? 

Reviewer #1: Yes

Reviewer #2: Yes

4. Have the authors made all data underlying the findings in their manuscript fully available?

Reviewer #1: Yes

Reviewer #2: Yes

5. Is the manuscript presented in an intelligible fashion and written in standard English?

Reviewer #1: Yes

Reviewer #2: Yes

6. Review Comments to the Author

Reviewer #1: Thank you for your revisions. Your manuscript has improved significantly in terms of theoretical depth, methodological rigor, and clarity. You have effectively addressed most of the previous concerns. However, I still have some concerns that require minor revisions before the paper can be accepted. Below are the key points that should be addressed to improve the clarity and consistency of your manuscript.

1. Inconsistency in Data Description (Section 3.1 vs. Section 4.3.2)

Section 3.1 describes the dataset as 2018-2019, while Section 4.3.2 states that 2023-2024 data was collected for robustness checks. This inconsistency must be resolved. Revise Section 3.1 to acknowledge the use of 2023-2024 data for validation, ensuring clarity on dataset scope.

2. Justification for 2023-2024 Data Inclusion

The manuscript does not explicitly state that 2023-2024 data was included to address concerns about data timeliness. Clarify in Section 3.1 that this dataset was collected as the latest available data to validate the findings. In Section 4.3.2, briefly discuss whether market conditions, platform policies, or user behavior have changed, and if not, explicitly state that the dataset remains comparable.

3. Comparison Between 2018-2019 and 2023-2024 Data

The robustness test confirms that findings remain stable, but the manuscript does not compare key variable distributions between the two datasets. Provide a brief descriptive analysis of whether variables such as negative reviews, app performance, and innovation frequency have changed significantly over time. If no substantial differences exist, explicitly state this to reinforce the consistency and validity of the results.

Reviewer #2: Thank you for your extensive, thoughtful and high-quality responses to the reviewer’s comments, and I am satisfied with all of the points raised.

7. PLOS authors have the option to publish the peer review history of their article (what does this mean? ). If published, this will include your full peer review and any attached files.

**Do you want your identity to be public for this peer review?** For information about this choice, including consent withdrawal, please see our Privacy Policy .

Reviewer #1: No

Reviewer #2: No

---

## [Author Response · Author response to Decision Letter 2]

2 Apr 2025

Thank you very much for your great comments and suggestions on our manuscript. We are so encouraged that you think the manuscript has improved significantly in terms of theoretical depth, methodological rigor, and clarity. We have carefully studied the comments from you and tried our best to respond.

Below we provide a point-to-point response to your comments:

1. Inconsistency in Data Description (Section 3.1 vs. Section 4.3.2)

Section 3.1 describes the dataset as 2018-2019, while Section 4.3.2 states that 2023-2024 data was collected for robustness checks. This inconsistency must be resolved. Revise Section 3.1 to acknowledge the use of 2023-2024 data for validation, ensuring clarity on dataset scope.

Response: Thank you for the constructive suggestion! Following your suggestion, we have added the description to acknowledge the use of 2023-2024 data for robustness checks in Section 3.1 (see, the lines marked in red on Page 9).

2. Justification for 2023-2024 Data Inclusion

The manuscript does not explicitly state that 2023-2024 data was included to address concerns about data timeliness. Clarify in Section 3.1 that this dataset was collected as the latest available data to validate the findings. In Section 4.3.2, briefly discuss whether market conditions, platform policies, or user behavior have changed, and if not, explicitly state that the dataset remains comparable.

Response: This is a really great suggestion and thank you very much! In this revision, following your suggestion, firstly, we have demonstrated that the data in 2023-2024 is collected as latest available data to address concerns about data timeliness ( see, the 1-3 lines of second paragraph in the subsection “4.3.2 Robustness Tests” on Page 14).

Secondly, we discuss the difference of market conditions, platform policies and user behavior between 2023-2024 and 2018-2019. The results show that the market environment in which apps operate has experienced more intense competition in 2023-2024. Continuous innovation has become increasingly crucial for developers, reflecting that the two sets of data from different time periods are comparable. ( see, the 3-8 lines of second paragraph in the subsection “4.3.2 Robustness Tests” on Page 14).

3. Comparison Between 2018-2019 and 2023-2024 Data

The robustness test confirms that findings remain stable, but the manuscript does not compare key variable distributions between the two datasets. Provide a brief descriptive analysis of whether variables such as negative reviews, app performance, and innovation frequency have changed significantly over time. If no substantial differences exist, explicitly state this to reinforce the consistency and validity of the results.

Response: Thanks for pointing this out! Following your suggestion, this paper presents the descriptive statistics of the latest data in table 4. Specifically, we add the descriptive analysis to compare the differences between 2018-2019 and 2023-2024 data. There are no significant or substantial differences between the two datasets ( see, the third paragraph in the subsection “4.3.2 Robustness Tests” on Page 14).

Additionally, his paper has used the two-way fixed effects model has been used to test the hypotheses, effectively controlling for the influence of time factors in the regression analysis in table 2. The results of 2018-2019 data have validity. ( see, the subsection “3.2.5 Model Specification” and table 2; the 4-6 lines of fourth paragraph in the subsection “4.3.2 Robustness Tests” on Page 15).

---

## [Editor Report · Decision Letter 2]

4 Apr 2025

Impact of online negative and positive reviews on App Innovation

PONE-D-24-42585R2

Dear Dr. Fu,

We’re pleased to inform you that your manuscript has been judged scientifically suitable for publication and will be formally accepted for publication once it meets all outstanding technical requirements.

Kind regards,

Ayesha Maqbool, PhD

Academic Editor

PLOS ONE
---

## [Editor Report · Acceptance letter]

PONE-D-24-42585R2

PLOS ONE

Dear Dr. Fu,

I'm pleased to inform you that your manuscript has been deemed suitable for publication in PLOS ONE. Congratulations! Your manuscript is now being handed over to our production team.

Kind regards,

on behalf of

Dr. Ayesha Maqbool

Academic Editor

PLOS ONE